# Quantitative spatial and temporal assessment of regulatory element activity in zebrafish

**Shipra Bhatia[1]\*, Dirk Jan Kleinjan[2†], Kirsty Uttley[1†], Anita Mann[1], Nefeli Dellepiane[1], Wendy A Bickmore[1]**

[1]MRC Human Genetics Unit, Institute of Genetics & Cancer, University of Edinburgh, Edinburgh, United Kingdom; [2]Centre for Mammalian Synthetic Biology at the Institute of Quantitative Biology, Biochemistry, and Biotechnology, SynthSys, School of Biological Sciences, University of Edinburgh, Edinburgh, United Kingdom

**Abstract** Mutations or genetic variation in noncoding regions of the genome harbouring cis-regulatory elements (CREs), or enhancers, have been widely implicated in human disease and disease risk. However, our ability to assay the impact of these DNA sequence changes on enhancer activity is currently very limited because of the need to assay these elements in an appropriate biological context. Here, we describe a method for simultaneous quantitative assessment of the spatial and temporal activity of wild-type and disease-associated mutant human CRE alleles using live imaging in zebrafish embryonic development. We generated transgenic lines harbouring a dual-CRE dual-reporter cassette in a pre-defined neutral docking site in the zebrafish genome. The activity of each CRE allele is reported via expression of a specific fluorescent reporter, allowing simultaneous visualisation of where and when in development the wild-type allele is active and how this activity is altered by mutation.

**\*For correspondence:**
shipra.bhatia@ed.ac.uk

[†]These authors contributed equally to this work

**Competing interest:** The authors declare that no competing interests exist.

## Introduction

Mutations or single-nucleotide polymorphisms (SNPs) in noncoding regions of the human genome functioning as cis-regulatory elements (CREs) or enhancers have been widely implicated in human disease and disease predisposition (*Bhatia and Kleinjan, 2014*; *Chatterjee and Ahituv, 2017*). Disease-associated sequence variation in enhancers can alter transcription factor (TF) binding sites, leading to aberrant enhancer function and altered target gene expression (*Bhatia and Kleinjan, 2014*). Next-generation sequencing technologies combined with molecular genetic approaches have enabled widespread identification of presumptive CREs and associated putative pathogenic mutations in patient cohorts (*Ryan and Farley, 2020*). However, compared to coding regions where the functional consequence of genetic variants can be extrapolated from knowledge about protein's structure and function, incomplete understanding of the TF binding potential of CREs impedes functional assessment of pathogenicity of genetic variants in the noncoding genome. Thus, determining how mutations in the vast stretches of the human noncoding genome contribute to disease and disease predisposition remains a huge unmet challenge.

Functional analysis of enhancer activity, and assessing the impact of disease-associated variation on this activity, depends on the availability of the right TFs in the right stoichiometric concentrations, which is only precisely captured in vivo. Enhancer-reporter transgenic assays have been widely employed in a variety of model organisms, including the mouse, to assess enhancer function in vivo (*Ashery-Padan and Gruss, 2001*; *Bhatia et al., 2015*; *Farley et al., 2015*; *Rogers and Williams, 2011*; *Visel et al., 2007*). These assays however can be affected by the random integration of transgenes and have

limited application for studying the temporal aspects of enhancer function over the time course of embryonic development since, for example, live imaging is challenging due to the opaqueness of the mouse embryo and its in utero embryonic development.

Zebrafish (*Danio rerio*) is a highly suitable in vivo vertebrate model for visualising tissue-specific enhancer activity. Robust transgenesis methods allow rapid generation of transgenic lines yielding transparent embryos which develop externally (*Mann and Bhatia, 2019*; *Phillips and Westerfield, 2014*). The activities of a large number of putative human and mouse CREs have been assessed in transgenic zebrafish models, irrespective of the primary sequence conservation of the mammalian CREs in zebrafish (*Bhatia et al., 2013*; *Bhatia et al., 2015*; *Chahal et al., 2019*; *Goode and Elgar, 2013*; *Rainger et al., 2014*; *Ravi et al., 2013*; *Yuan et al., 2018*). However, these assays were based on Tol2 recombination which mediates random integration of the CRE-reporter cassette in the zebrafish genome (*Kawakami et al., 2000*). The measured CRE activities were strongly influenced by the variable site and copy number of integrations, necessitating analysis of each element in multiple transgenic lines and precluding quantitative assessment of CRE activities. These biases can be alleviated by targeted integration of the transgenic cassette into pre-defined neutral sites in the zebrafish genome using phiC31-mediated recombination (*Hadzhiev et al., 2016*; *Mosimann et al., 2013*; *Roberts et al., 2014*).

Previously, we developed a system in which dual fluorescence CRE-reporter zebrafish transgenics allow for direct comparison of the in vivo spatial and temporal activity of wild-type (Wt) and putative SNP/mutation (Mut) bearing CREs in the same developing embryo (*Bhatia et al., 2015*). The functional output from each CRE version (Wt/Mut) is visualised simultaneously as eGFP or mCherry signal within a live developing embryo bearing both transgenes. This enables unambiguous comparison of the activity of both Wt and mutant CREs in a developmental context, simultaneous assessment of multiple separate elements for subtle differences in spatio-temporal overlap, and the validation of putative TFs by analysing the effect of morpholino-mediated depletion of the putative TF on CRE activity (*Bhatia et al., 2015*). The assay had clear advantages over other conventional CRE-reporter transgenic assays, notably rapid, unambiguous detection of subtle differences in CRE activities using a very low number of animals. However, as the CRE alleles were on separate constructs randomly integrated into the zebrafish genome, the assay was not suitable for quantitative assessment of altered CRE activity. Furthermore, multiple transgenic lines had to be analysed for each CRE to eliminate any bias arising from the site of integration.

Here we describe Q-STARZ (Quantitative Spatial and Temporal Assessment of Regulatory element activity in Zebrafish), a new and significantly improved design of our previous transgenic reporter assay, based upon targeted integration of a dual-CRE dual-reporter cassette into a pre-defined site in the zebrafish genome (*Figure 1*). A unique feature of this design is the single transgenic cassette containing both Wt and mutant CREs, separated by strong insulator sequences, with the transcriptional potential of both CREs read out as expression of different fluorescent proteins. Qualitative and quantitative activity of the two CRE alleles is analysed from eGFP/mCherry fluorescence in real time by live imaging of embryos obtained from the founder (F0) lines bearing the dual CRE dual-reporter cassette. This allows robust, unbiased assessment of spatial and temporal activities of both CREs using a single transgenic line, thereby reducing animal usage by up to 75% compared to the previous design. We utilise disease-associated mutations in well-characterised CREs from the *PAX6* and *SHH* loci to demonstrate the salient features of the Q-STARZ method.

## Results

### Targeted integration of a dual-CRE dual-reporter transgenic cassette in the zebrafish genome

Analysis of enhancer activities in conventional zebrafish reporter assays suffers from bias arising from position effects due to the random integration of the transgene at Tol2 sites naturally distributed at a low frequency throughout the zebrafish genome (*Kawakami et al., 2000*; *Mann and Bhatia, 2019*). Most assay designs also harbour only one CRE per transgene introducing ambiguity in the analysis when comparing CREs with highly similar activities or subtle changes in sequence (e.g. disease-associated mutations or SNPs). Q-STARZ is a versatile, robust and cost-effective analysis pipeline designed to alleviate both these limitations (*Figure 1*).

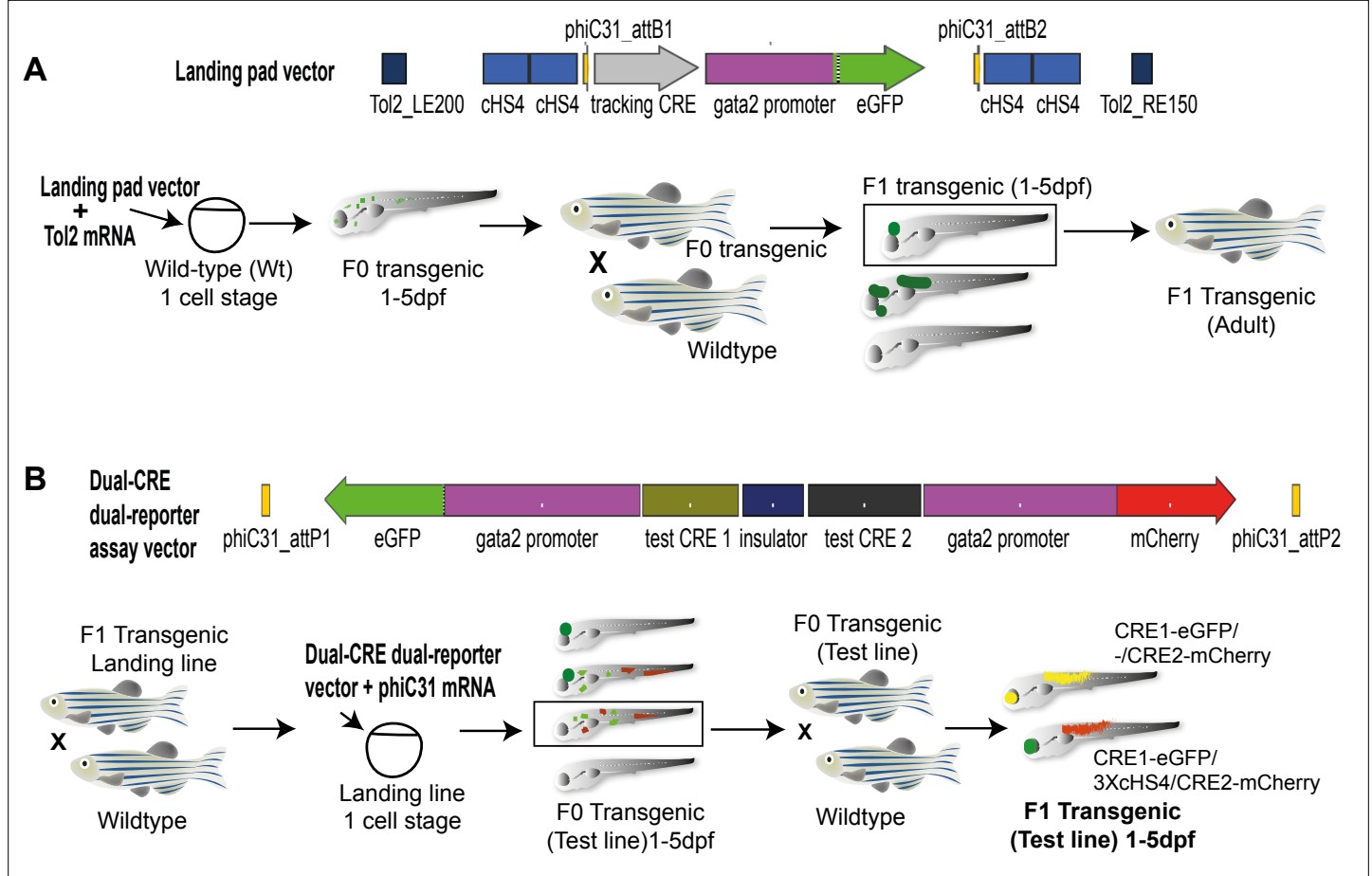

**Figure 1.** Quantitative Spatial and Temporal Assessment of Regulatory element activity in Zebrafish (Q-STARZ) pipeline. Diagramatic representation of the Q-STARZ pipeline. (**A**). Top: map of the landing pad vector. Bottom: scheme for generating stable transgenic 'landing lines'. The landing pad vector is co-injected with Tol2 mRNA into one-cell stage wild-type embryos. Tol2-mediated recombination integrates the landing pad containing phiC31-attB sites flanking the tracking cis-regulatory element (CRE)-reporter cassette (*SHH*-SBE2, a CRE driving eGFP in the developing forebrain) at random locations in the zebrafish genome. F0 embryos showing mosaic eGFP expression are raised to adulthood. F1 embryos obtained by outcrossing F0 lines with wild-type zebrafish are screened for tracking CRE-driven reporter (eGFP) expression. Embryos where eGFP expression was only observed in the expected activity domain (forebrain) of the tracking CRE were raised to adulthood to establish stable 'landing lines'. (**B**) Top: map of the dual-CRE dual-reporter vector. Bottom: scheme for replacing the tracking cassette in the landing line with the dual-CRE dual-reporter cassette containing the enhancers to be assayed for spatiotemporal activity. Assay vector and mRNA coding for phiC31 integrase are injected in one cell stage embryos derived from outcrossing F1 landing line with wild-type fish. Injected embryos were selected for loss of tracking CRE (*SHH*-SBE2)-driven eGFP fluorescence in forebrain and mosaic expression of both eGFP and mCherry resulting from the test CREs in the assay cassette. F0 transgenic lines were established from selected embryos and eGFP and mCherry expression imaged in F1 embryos derived from outcrossing these lines with wild-type fish. Signals from both reporters were observed in the activity domains of both CREs in F1 embryos bearing the assay constructs with 'neutral' sequence between the two CRE-reporter units (yellow signal seen in expressing tissues in the merge channel). However, eGFP and mCherry expression was restricted to tissues where the associated CREs are active upon inclusion of three copies of the chicken β-globin 5'HS4 (3XcHS4) insulator between the two CRE-reporter units.

The online version of this article includes the following figure supplement(s) for figure 1:

**Figure supplement 1.** Diagrammatic representation of the gateway cloning strategy used for generating the landing pad (A) and dual-cis-regulatory element (CRE) dual-reporter vector (B).

We first generated 'landing lines' harbouring phiC31 attB integration sites at inert positions in the zebrafish genome. Using Tol2-mediated transgenesis, we integrated 'landing pads' at random sites in the zebrafish genome (*Figure 1A*, *Figure 1—figure supplement 1*). To visualise successful integration events, the landing pads contain 'tracking CREs' (*Supplementary file 1*) driving expression of a 'tracking reporter gene'. These CREs had previously well-characterised activities, enabling us to select transgenic lines devoid of bias arising from the site of integration (*Bhatia et al., 2015*). We assessed reporter gene expression in F1 embryos derived from several independent F0 transgenic lines for

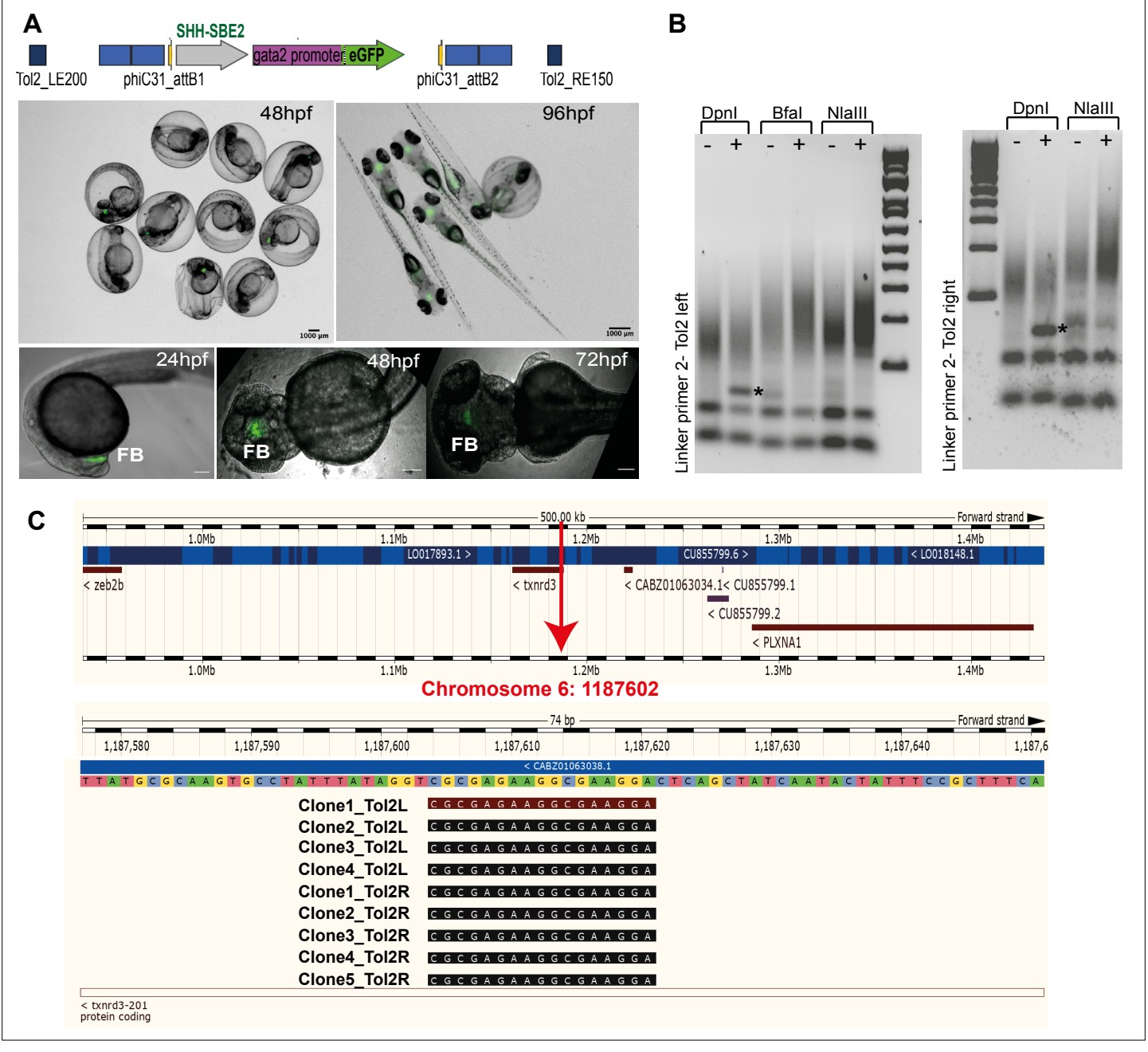

**Figure 2.** Characterisation of SHH-SBE2 landing line. (**A**) Top: schematic of the design of the landing pad bearing SHH-SBE2 as the tracking cis-regulatory element (CRE). Below: CRE activity observed exclusively in the forebrain in F1 embryos with the SHH-SBE2-eGFP tracking cassette. Images shown for pool of F1 embryos (scale bar = 1000 μm) and individual embryos (scale bar = 100 μm) at different stages of embryonic development. FB, forebrain; hpf, hr post fertilisation. Scale bar = 100 μm. (**B**) Unique bands (*) observed in round 2 of PCR amplification of DpnI digested genomic DNA from F1 embryos bearing the landing pad cassette. (**C**) Ensembl Genome Browser snapshot depicting the integration site (red arrow) of the SHH-SBE2 landing pad and sequencing data from clones bearing the PCR product shown by * in (**B**).

The online version of this article includes the following figure supplement(s) for figure 2:

**Figure supplement 1.** Landing lines with tracking cis-regulatory element (CRE)-driven reporter gene expression influenced by site of integration.

each tracking CRE (*Figure 2A*, *Figure 2—figure supplement 1*, *Supplementary file 2*). F1 embryos in which the activity of CRE was not influenced by the site of integration were raised to adulthood to establish 'landing lines' presumed to be harbouring the phiC31 attB sites in an inert position of the zebrafish genome (*Figure 1A*). CRE activities were observed to be highly influenced by the site

of integration in F1 embryos derived from founder lines bearing *SOX9*-CNEa and *Pax6*-SIMO CREs (*Figure 2*, *Figure 2—figure supplement 1*), but we obtained three independent landing lines with a clean eGFP expression pattern using *Shh*-SBE2 as the tracking CRE (*Supplementary file 2*). *Shh*-SBE2 is a forebrain enhancer driving *Shh* expression in the hypothalamus (*Jeong and Epstein, 2003*). Based on these observations, we decided to use the *Shh*-SBE2 landing line for all subsequent experiments described in this study. The precise integration site of the landing pad in the three *Shh*-SBE2 lines was determined using ligation-mediated PCR (LM-PCR) and transgene segregation analysis (described in Materials and methods). Based on these observations, we decided to use the *Shh*-SBE2 landing line with a clean single-site integration for all subsequent experiments described in this study (*Figure 2*).

In the second part of the Q-STARZ pipeline, we generated a 'dual-CRE dual-reporter assay construct' containing two CRE-reporter cassettes separated from each other by strong insulator sequences (*Figure 1B*, *Figure 1—figure supplement 1*). The assay construct was co-injected with mRNA encoding phiC31 integrase into F2 embryos derived from the *Shh*-SBE2 landing line. Recombination-mediated cassette exchange between the attB sites on the landing pad construct and attP sites on the assay construct integrates a single copy of the dual-CRE dual-reporter cassette at the pre-defined site in the zebrafish genome (*Figure 1B*). Injected embryos were scored for loss of *Shh*-SBE2-driven CRE activity in the forebrain and gain of mosaic eGFP and mCherry signals from the assay CRE-reporter cassette. These were scored as successful flipping events and were observed at a frequency of about 10% of the injected embryos. Selected embryos were raised to sexual maturity to establish 2–3 independent founder transgenic lines for each assay cassette analysed in this study (*Supplementary file 2*). Activity of both CREs was visualised simultaneously as eGFP or mCherry signals by live imaging of F1 embryos derived from outbreeding the founder lines with Wt zebrafish (*Figure 1B*). Detailed protocols for the various steps described in this section are provided in Materials and methods.

## Robust, quantitative assessment of CRE activity using Q-STARZ

A key feature of Q-STARZ is the simultaneous assessment of activities of the two CREs present on the assay cassette. In order to prevent crosstalk between the two enhancers, a well-characterised insulator sequence from the chicken genome, chicken β-globin 5'HS4 (cHS4) (*Chung et al., 1997*; *Wang et al., 1997*), was placed between the two CRE-reporter cassettes (*Figure 1B*, *Figure 1—figure supplement 1*). We optimised the assay using constructs bearing two CREs with previously well-characterised tissue-specific activities from the *PAX6* regulatory domain (*Figure 3*). PAX6 is a TF with vital pleiotropic roles in embryonic development (*Ashery-Padan and Gruss, 2001*; *Kleinjan and van Heyningen, 2005*; *Osumi et al., 2008*) and >30 CREs have been characterised which coordinate precise spatial and temporal *PAX6* expression in the developing eyes, brain and pancreas (*Bhatia and Kleinjan, 2014*). We selected *PAX6*-7CE3 and *PAX6*-SIMO for this analysis as they have well-established and highly distinct tissue-specific activities during zebrafish embryogenesis. *PAX6*-7CE3 drives expression in the hindbrain and neural tube from 24 to 120 hr post fertilisation (hpf), while *PAX6*-SIMO activity is in developing lens and forebrain from 48 to 120 hpf (*Bhatia et al., 2013*; *Ravi et al., 2013*; *Supplementary file 1*).

When the two CRE-reporter cassettes were separated by a 'neutral' sequence – a randomly selected region from the mouse genome with no insulator activity – we observed complete crosstalk of the two CRE activities (*Figure 3A*, *Figure 3—figure supplement 1*, *Supplementary file 2*). We also performed a dye-swap experiment wherein the eGFP and mCherry reporters were swapped between the two CREs. We observed no significant difference in CRE activities in the dye-swap experiment, indicating no bias was introduced by varying signal intensities from the two fluorophores used (*Figure 3A*, *Figure 3—figure supplement 1*). Next, we substituted the neutral sequence with one, two or three tandem copies of the cHS4 insulator. Enhancer blocking activity of this insulator has been attributed to its ability to bind CTCF (*Bell et al., 1999*). Crosstalk between the two enhancer-reporter cassettes was progressively reduced with increasing copies of cHS4, with complete insulation achieved in replacement cassettes bearing three copies (3xcHS4) (*Figure 3A*, *Figure 3—figure supplements 2–4*). We quantified the effects of the presence of insulator sequences by measuring eGFP and mCherry intensities in the expressing tissues at all stages of embryonic development in multiple embryos for each of the constructs analysed. Quantification was focussed on lens and hindbrain tissues as we observed expression at these sites consistently in all the lines analysed (*Supplementary file 2*). This analysis confirmed that, as the number of copies of the insulator increases, there

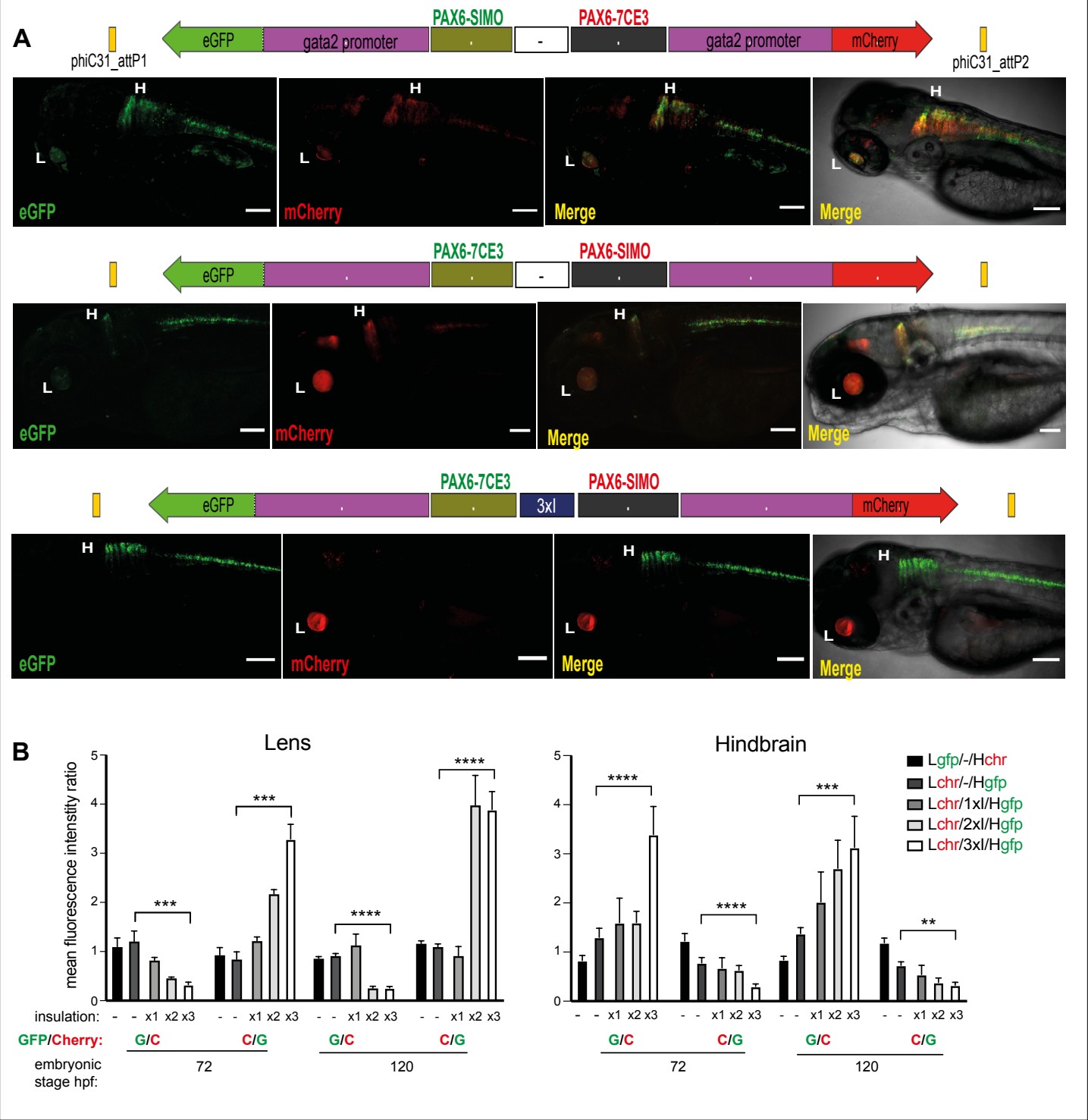

**Figure 3.** Quantitative assessment of tissue-specific enhancer activity and effect of insulation on crosstalk between cis-regulatory elements (CREs) in dual-CRE dual-reporter constructs. Constructs carrying well-characterised CREs from the PAX6 locus (*PAX6*-7CE3, hindbrain enhancer, and *PAX6*-SIMO, lens enhancer). (**A**) Confocal images of 96hpf F1 embryos derived from founder lines injected with the cassettes indicated above each image panel. Top two panels show dye-swap experiment (eGFP and mCherry reporters swapped between the two CREs) with a neutral sequence (–, no insulator activity) between the two CRE-reporter cassettes. eGFP and mCherry expression is observed in both lens and hindbrain indicating complete crosstalk between 7CE3 and SIMO CREs. Bottom panel: inclusion of three copies of the well-characterised chicken β-globin 5′HS4 (3XcHS4) insulator restricts the activities of each enhancer to their respective specific domains. Scale bars = 100 µm. (**B**) Average of mean fluorescence intensities ratios (G/C: eGFP/mCherry; C/G: mCherry/eGFP) in the lens and hindbrain at 72 and 120 hr post fertilisation (hpf) in F1 embryos derived from founders bearing constructs without

*Figure 3 continued on next page*

*Figure 3 continued*

(–) or with 1×, 2× or 3× insulator sequences. Each bar indicates average of ratios of mean fluorescence intensities from at least five independent images of embryos bearing the assay construct indicated (n ≥ 5, error bars indicate standard deviation). A highly significant difference in fluorescence intensity ratios (computed by two-tailed Student's t-test) was observed between embryos at the same stage of development harbouring constructs with no insulator (–) and those with three copies of the insulator (3xI). This demonstrates that fluorescence is progressively restricted to the tissue where the associated CRE is active as the number of copies of the insulator increases. Raw data used for plotting the graphs are provided in *Figure 3—source data 1*. L, Lens; H, hindbrain, ****p<0.0001, ***p<0.001, **p<0.01.

The online version of this article includes the following figure supplement(s) for figure 3:

**Source data 1.** Quantification data of eGFP and mCherry intensities in transgenic lines bearing the assay constructs described in *Figure 3*.

**Figure supplement 1.** Assessment of tissue-specific cis-regulatory element (CRE) activity from dual-CRE dual-reporter constructs with neutral sequence between CREs.

**Figure supplement 2.** Assessment of tissue-specific cis-regulatory element (CRE) activity from the dual-CRE dual-reporter replacement construct with one copy of insulator sequence.

**Figure supplement 3.** Assessment of tissue-specific cis-regulatory element (CRE) activity from the dual-CRE dual-reporter replacement construct with two copies of insulator sequence.

**Figure supplement 4.** Assessment of tissue-specific cis-regulatory element (CRE) activity from the dual-CRE dual-reporter replacement construct with three copies of insulator sequence.

is progressively restricted expression of the reporters towards expression only in the activity domains of their associated CRE (*Figure 3B*).

## Dissecting spatial and temporal dynamics of CREs with highly overlapping activities using live imaging

A salient feature of Q-STARZ is the ability to simultaneously visualise the activity of both CREs on the assay cassette in the same developing zebrafish embryo in real time using live imaging. To establish proof of principle, we investigated the precise spatial and temporal activities of two CREs from the *Shh* locus, *Shh*-SBE2 and *Shh*-SBE4, previously demonstrated to have highly similar domains of activity in the developing forebrain of mouse embryos (*Figure 4*; *Jeong and Epstein, 2003*; *Jeong et al., 2008*). We analysed the activities of the two CREs in transgenic lines generated with two assay constructs (*Shh*-SBE2-eGFP/3xcHS4/*Shh*-SBE4-mCherry and *Shh*-SBE2-mCherry/3xcHS4/*Shh*-SBE4-eGFP) to avoid any bias arising from stability of the fluorophores. Our analyses revealed unique, as well as overlapping, domains of activity of both CREs in the early stages of forebrain development (~24–50 hpf) (*Figure 4*, *Video 1*). However, from ~60 to 120 hpf, the activities of both CREs are in completely distinct domains of the developing forebrain with no overlapping activity observed. *Shh*-SBE2 was active in the rostral part of forebrain while *Shh*-SBE4 activity was restricted to caudal forebrain (*Figure 4*, *Video 2*). This analysis highlights the importance of simultaneous visualisation of CRE activities in the developing embryo to define the precise spatial and temporal activity of each CRE.

## Robust assessment of the effects of disease-associated mutations on CRE activity

As well as qualitative comparison of activity between two different CREs, a key strength of the Q-STARZ pipeline is its suitability for discerning the precise effects of disease-associated mutations or SNPs within a specific CRE. We tested this in SBE2, a regulatory element that controls *SHH* expression in the developing forebrain, using a point mutation (C>T) identified in a patient with holoprosencephaly and shown to abrogate the activity of SBE2 in the rostral hypothalamus of the mouse (*Figure 5*; *Bhatia et al., 2015*; *Jeong et al., 2008*). We simultaneously visualised the activities of the human Wt(C) and Mut(T) SBE2 alleles in our dual-CRE dual-reporter system by live imaging of transgenic zebrafish embryos from 24 to 72 hpf (SBE2-Wt(C)-eGFP/3xcHS4/SBE2-Mut(T)-mCherry and SBE2-Wt(C)-mCherry/3xcHS4/SBE2-Mut(T)-eGFP, *Figure 5*, *Video 3*). We detected no difference in the activities of the two alleles in very early development until ~40 hpf. However, from ~48 to 72 hpf, activity of the alleles started to diverge. Expression driven by the Wt allele was observed in the developing rostral and caudal hypothalamus of transgenic embryos while the Mut allele was only active in the caudal hypothalamus, indicating that the mutation disrupts rostral activity of the SBE2. Upon quantification of reporter gene expression associated with each allele, we observed no significant

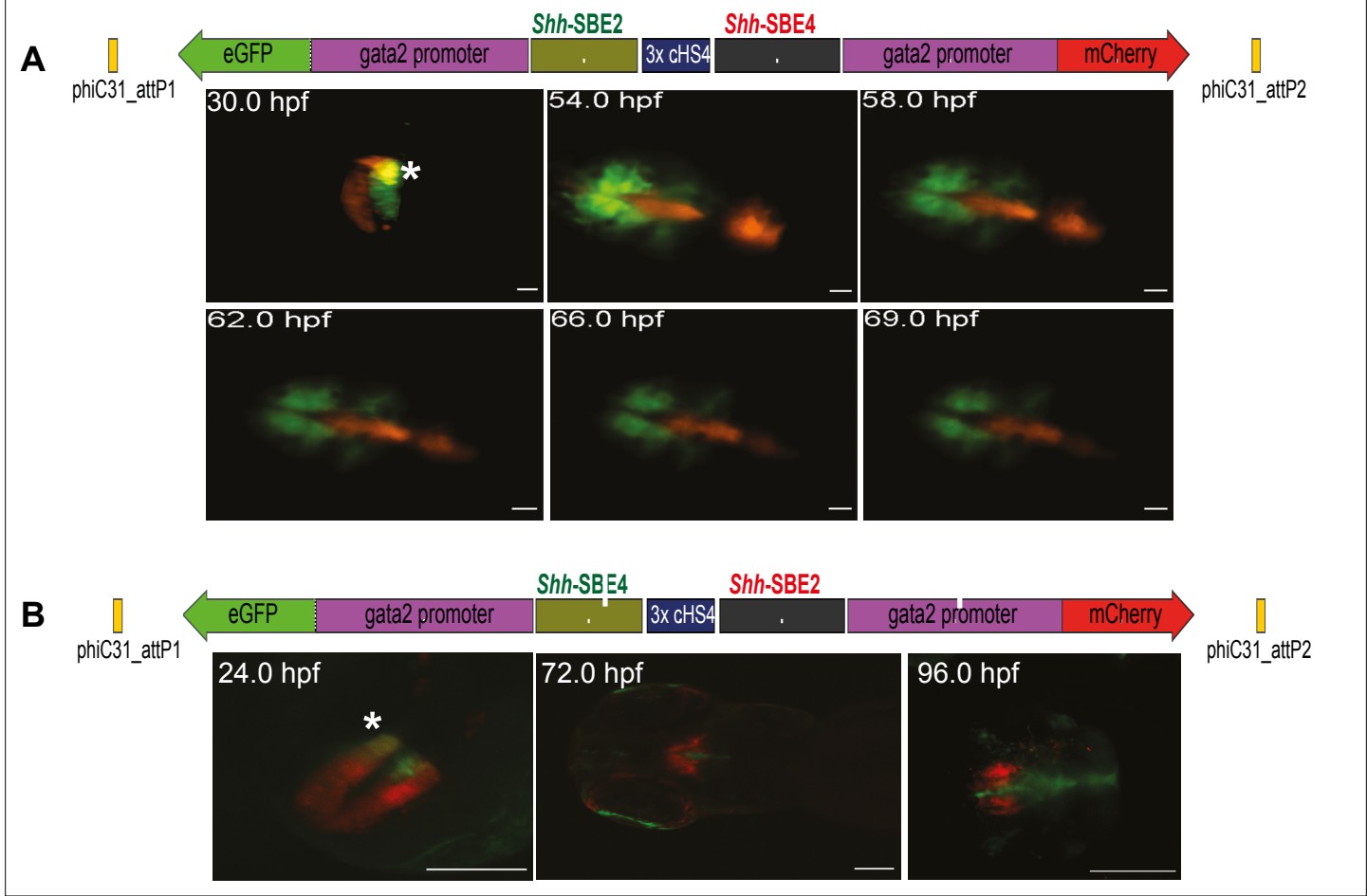

**Figure 4.** Live imaging of transgenic embryos to detect subtle differences in spatial and temporal enhancer activities. (**A**) Top: schematic of assay construct with two enhancers from the mouse *Shh* locus active in developing forebrain (*Shh*-SBE2 and *Shh*-SBE4 driving eGFP and mCherry respectively). Below: snapshots of live imaging of F1 embryos derived from transgenic lines bearing the assay construct. Distinct as well as overlapping domains (marked by *) of activities are observed for the two cis-regulatory elements (CREs) in early stages of development, until about 54 hr post fertilisation (hpf). At later stages of embryonic development, the activities of the two forebrain CREs are observed in completely distinct domains. Scale bar = 100 μm. (**B**) As in (**A**) but with a dye-swap, that is, *Shh*-SBE2 driving mCherry and *Shh*-SBE4 driving eGFP.

The online version of this article includes the following figure supplement(s) for figure 4:

**Figure supplement 1.** Genotyping assay for assessing successful integration of the replacement construct in the landing sites.

difference in activity between the two alleles at 28 hpf. However, at later stages of development (48 and 72 hpf), the mutant allele failed to drive reporter gene expression in the rostral hypothalamus and had significantly weaker activity in the caudal hypothalamus (*Figure 5*). Our analysis thus unambiguously and precisely uncovered where and when in embryonic development the mutation associated with holoprosencephaly alters the enhancer activity of SBE2.

## Discussion

The noncoding region of the human genome is estimated to contain approximately 1 million enhancers (*Consortium, 2012*; *Thurman et al., 2012*). The widespread application of whole-genome sequencing for understanding genetic diseases (rare, common and acquired – i.e. cancer), combined with genome-wide identification of chromatin signatures associated with active enhancers, has led to the identification of a large number of putative enhancers with disease-associated or disease risk-associated sequence variation (*Bhatia and Kleinjan, 2014*; *Chatterjee and Ahituv, 2017*; *Short et al., 2018*; *Wu and Pan, 2018*). A complete understanding of how these sequence changes alter enhancer function is a necessary first step towards establishing roles of the CREs in the aetiology of the associated disease.

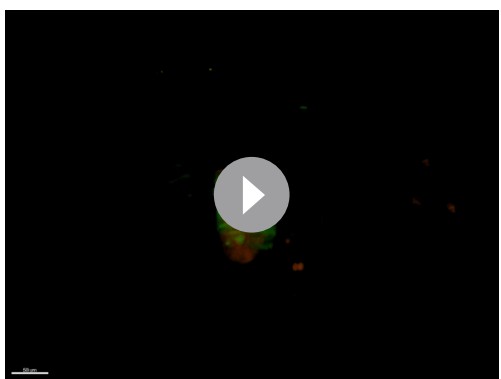

**Video 1.** Confocal imaging of 30 hr post fertilisation embryo derived from transgenic line bearing the *Shh*-SBE2gfp/3XcHS4/*Shh*-SBE4mCherry assay construct. The distinct expression domains of SBE2 and SBE4 enhancers in the developing forebrain are seen in green and red, respectively, while the region where their activities overlap is depicted in yellow.
https://elifesciences.org/articles/65601/figures#video1

Thus, there is a pressing need for rapid, cost-effective assays for robust unambiguous comparisons of mutant CRE alleles with the activities of Wt alleles. Importantly, this has to be done in the appropriate context, relevant to the biology of the associated disease. CRE activity depends on precise stoichiometric concentrations of specific TFs, which is only achieved in the right physiological context inside a developing embryo or in cell lines that closely model the cellular phenotypes of developing tissues (*Sasai et al., 2012*; *Weedon et al., 2014*).

The Q-STARZ assay we describe here is highly versatile and enables unambiguous assessment of human tissue-specific CRE function in vivo at all stages of early embryonic development in a vertebrate model system. A distinctive feature of the assay is the targeted integration of a single transgenic cassette bearing two independent CRE-reporter units into a pre-defined inert site in the zebrafish genome. We establish an analysis pipeline that enables simultaneous robust qualitative and quantitative analysis of enhancer function without any bias from position effects or copy number variation between the two CREs analysed.

Similar methods of targeted integration of enhancer-reporter transgenic cassettes have been developed for zebrafish as well as mouse models (*Kvon et al., 2020*; *Mosimann et al., 2013*). Q-STARZ however offers a unique advantage when analysing the effects of disease-associated sequence variation on CRE function by enabling direct comparisons of activities of Wt and mutant alleles inside the same, transparent developing embryo using live imaging. Docking the dual-CRE dual-reporter cassette into a pre-defined site in the zebrafish genome ensures no variability in transgene expression patterns.

Using CREs with previously well-established activities, we demonstrate that inclusion of three tandem copies of a strong insulator sequence in the a construct robustly prevents crosstalk between the two CREs analysed. This feature enables direct comparisons of the spatial and temporal dynamics of both CREs by simultaneous visualisation of functional outputs in live embryos at all stages of development. We have convincingly demonstrated that the activities of the two CREs tested in the assay cassette can be shielded from each other by including three copies of the cHS4 insulator in the cassette. However, if imperfect shielding is observed for any CRE pairs the assay can be adapted to use higher copies of the cHS4 insulator or other sequences with demonstrated insulator function, for example, FB insulator (*Ramezani et al., 2008*). We have also rigorously employed dye-swap experiments in this article to demonstrate CRE activities in our assay are not biased by the choice of

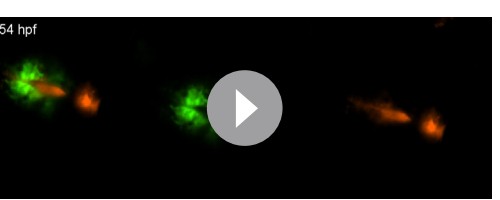

**Video 2.** Time-lapse video of embryo derived from transgenic line bearing the *Shh*-SBE2gfp/3XcHS4/*Shh*-SBE4mCherry assay construct. Images were acquired from 54 to 69 hr post fertilisation, with a time interval of 1 hr. The distinct expression domains of SBE2 and SBE4 enhancers in the developing forebrain are seen in green and red, respectively.
https://elifesciences.org/articles/65601/figures#video2

fluorophores. However, we would endeavour to employ de-stabilised fluorophores, for example, dsRed (*Rodrigues et al., 2001*) in future iterations of our assay. We demonstrate here that this can uncover the precise sites and time points in embryonic development where the CRE functions are unique and where they overlap with each other. Q-STARZ is therefore an ideal tool for generating a detailed cell-type-specific view of CRE usage during embryonic development. This will enhance understanding of the roles of CREs in target gene regulation, particularly for the complex regulatory landscapes of genes with key roles in development like *PAX6* and *SHH*. Analysis

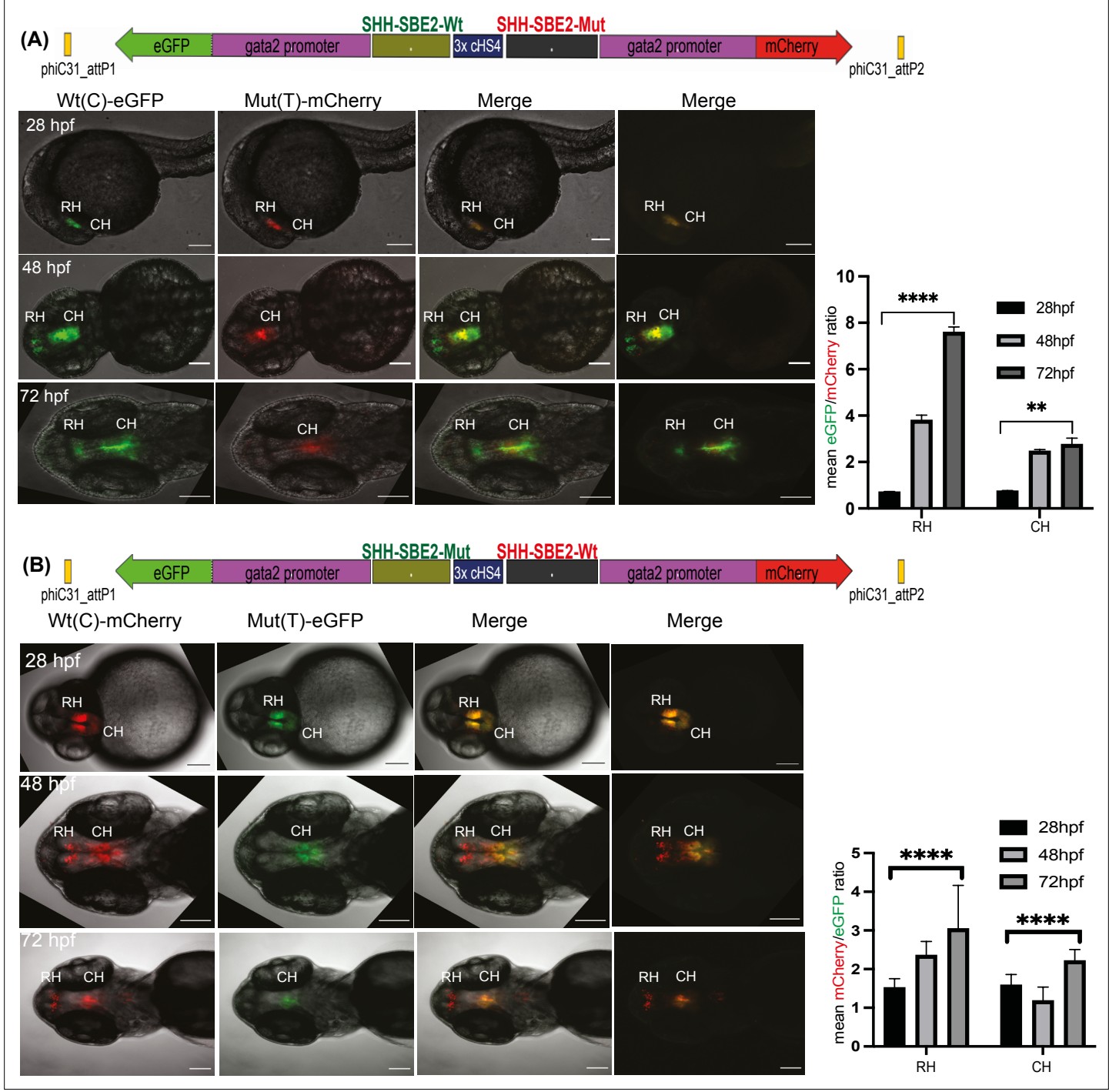

**Figure 5.** Quantitative assessment of altered CRE activity by disease-associated sequence variation. Dye swap experiment with SHH-SBE2 enhancer wild-type Wt(C) allele and Mut(T) allele bearing a holoprosencepaly- associated mutation (A: Wt(C)-eGFP/ Mut(T)-mCherry; B: Wt(C)-mcherry/ Mut(T)-eGFP).Top: Schematic representation of the test construct indicating the reporter genes driven by the two alleles. Bottom: confocal images and histogram of average of mean fluorescence intensities ratio (A:eGFP/mCherry; B:mcherry/eGFP) in the rostral (RH) and caudal (CH) hypothalamus for F1 embryos derived from founder lines bearing the test construct described (n≥5, error bars indicate standard deviation). At earlier stages of development (28-36 hpf, no significant difference in activity was observed between the two alleles). However at later stages of development (48 hpf and 72h pf) the mutant allele failed to drive reporter gene expression in the RH, and had significantly weaker activity in the CH at 72 hpf. Raw data used for plotting the graphs is provided in *Figure 5—source data 1*. ****p<0.0001, **p<0.01 (computed by two-tail student t-test). Scale bar = 100 μm.

The online version of this article includes the following figure supplement(s) for figure 5:

**Source data 1.** Quantification data of eGFP and mCherry intensities in transgenic lines bearing the assay constructs described in *Figure 5*.

**Video 3.** Time-lapse video of embryo derived from transgenic line bearing the *SHH*-SBE2-Wtgfp/3XcHS4/*SHH*-SBE2-Mut-mCherry assay construct. Images were acquired from 40 to 60 hr post fertilisation, with a time interval of 2 hr.

https://elifesciences.org/articles/65601/figures#video3

of CREs derived from these loci in conventional transgenic assays has revealed multiple CREs apparently driving target gene expression in the same or highly overlapping tissues and cell types. This has led to the concept of redundancy in enhancer function conferring robustness of expression upon genes with key roles in embryonic development (*Cannavò et al., 2016*; *Frankel et al., 2010*; *Osterwalder et al., 2018*). However, our analysis of the *Shh*-SBE2 and SBE4 enhancers, previously reported as forebrain enhancers with overlapping functions, reveals subtly distinct spatial and temporal activity domains of each enhancer during development. Based on these results, we hypothesise that there are small but important differences in the timing of action or precise localisation in cell types within the forebrain where these enhancers exert their roles that are overlooked when analysed independently in conventional transgenic assays.

Finally, we demonstrate that Q-STARZ can robustly detect differences in activities of mutant and Wt CRE alleles. Live imaging of transgenic embryos carrying a reporter cassette with a previously validated disease-associated point mutation in the *SHH*-SBE2 enhancer revealed the loss of activity of the mutant allele in the rostral hypothalamus compared to the Wt CRE. This recapitulates a similar loss of rostral activity of the SBE2 mutant that has been previously reported in mouse transgenic assays (*Jeong et al., 2008*). However, since we could visualise the activities of both the Wt and mutant alleles simultaneously in the same embryos in real time, we were able to determine the precise time point in development when the mutation affects CRE function. We propose that Q-STARZ will be a powerful tool to define the precise cell types and stages of development where CRE function is affected by mutations or SNPs identified by GWAS and other studies, thus this could significantly improve our ability to discern potentially pathogenic and functional sequence variation from background human genetic variation, which is currently a major challenge for human genetics. The analysis pipeline would only be suitable for CREs associated with genes active in early stages of embryonic development.

## Materials and methods

**Key resources table**

| Reagent type (species) or resource | Designation | Source or reference | Identifiers | Additional information |
|---|---|---|---|---|
| Commercial assay or kit | Gateway recombination cloning system | Invitrogen | 12535-019 | |
| Commercial assay or kit | Phusion high fidelity polymerase | NEB | M0530S | |
| Commercial assay or kit | TOPO TA Cloning Kit | Thermo Fischer Scientific | 451641 | |
| Commercial assay or kit | Plasmid purification columns | QIAGEN | 12123 | |
| Commercial assay or kit | PCR purification columns | QIAGEN | 28115 | |
| Commercial assay or kit | SP6 mMessage mMachine kit | Ambion | AM1340 | |
| Commercial assay or kit | DNeasy blood and tissue kit | QIAGEN | 69504 | |
| Commercial assay or kit | T4 ligase | NEB | M020S | |
| Recombinant DNA reagent | pCS2-TP (plasmid) | *Bischof et al., 2007* | | |
| Recombinant DNA reagent | pcDNA3.1 phiC31 (plasmid) | Addgene | Plasmid #68310 | |
| Recombinant DNA reagent | NlaIII (enzyme) | NEB | R0125S | |

*Continued on next page*

*Continued*

| Reagent type (species) or resource | Designation | Source or reference | Identifiers | Additional information |
|---|---|---|---|---|
| Recombinant DNA reagent | BfaI (enzyme) | NEB | R0568S | |
| Recombinant DNA reagent | DpnII (enzyme) | NEB | R0543S | |
| Chemical compound, drug | PTU (1-phenyl2-thio-urea) | Sigma-Aldrich | S515388 | |
| Chemical compound, drug | Low-melting point (LMP) agarose | Sigma-Aldrich | A9414 | |
| Chemical compound, drug | Tricaine | Sigma-Aldrich | MS222 | |
| Software, algorithm | Imaris | Bitplane, Oxford Instruments | RRID:SCR_007370 | |
| Software, algorithm | Fiji | | RRID:SCR_002285 | |
| Genetic reagent | *Danio rerio* | Strain AB | RRID: ZIRC_ZL1 | |

## Generation of landing pad and dual-CRE dual-reporter assay vectors

All the constructs in this study were generated using the Gateway recombination cloning system (Invitrogen). PCR primers with suitable recombination sites were used for amplification of CREs from the genomic DNA (*Supplementary file 1*). The PCR amplification was performed using Phusion high fidelity polymerase (NEB), and the amplified fragments were cloned in Gateway pDONR entry vectors (pP4P1r or pP2rP3) and sequenced using M13 forward and reverse primers for verification. The recombination sites attached in primers, entry vector for cloning and genomic DNA used in amplification for each CRE are indicated in *Supplementary file 1*. For generating the landing pad vector, pP4P1r entry vector with the tracking CRE and pDONR221 entry vector containing a gata2-eGFP (*Bhatia et al., 2015*) were recombined with a destination vector with a Gateway R4-R2 cassette flanked by phiC31_attB1/B2 and Tol2 recombination sites (*Figure 1*, *Figure 1—figure supplement 1*). The details of the tracking CREs are provided in *Supplementary file 1*. The assay vector was generated via three-way gateway reaction as described in *Figure 1*, *Figure 1—figure supplement 1*. The test CREs were cloned either in pP4P1r or pP2rP3 entry vectors and the insulator sequences and neutral sequence was cloned in pDONR221. For generating constructs with multiple copies of the insulator sequence, the sequences were first cloned in tandem in TOPO TA Cloning Kit (Thermo Fisher Scientific, cat no. 451641). Plasmids containing one, two or three copies of the insulator sequence were used as templates for amplification of products suitable for cloning in pDONR221. The destination vector was synthesised by Geneart and contained a Gateway R4-R3 cassette flanked by phiC31_attP1/P2 recombination sites and minimal promoter-reporter gene units (gata2-eGFP and gata2-mCherry). Gata2 promoter was used as the minimal promoter in both the landing pads and dual-CRE dual-reporter cassettes based on previous studies demonstrating robust promoter activity devoid of any basal level of reporter gene activation (*Bhatia et al., 2015*). Details of each construct generated in the article are provided in *Supplementary file 1*, and complete vector maps for all the constructs would be available on request.

## Generation of zebrafish transgenic lines

Zebrafish were maintained in a recirculating water system according to standard protocols (*Sprague et al., 2008*). Embryos were obtained by breeding adult fish of standard stains (AB, RRID:ZIRC_ZL1) and raised at 28.5°C as described (*Sprague et al., 2008*). Embryos were staged by hpf as described (*Kimmel et al., 1995*). Final CRE-reporter plasmids were isolated using QIAGEN miniprep columns and were further purified on a QIAGEN PCR purification column (QIAGEN) and diluted to 50 ng/ml with nuclease-free water. Tol2 transposase mRNA and phiC31 integrase mRNA were synthesised from a NotI-linearised pCS2-TP or pcDNA3.1 phiC31 plasmid, respectively (*Bischof et al., 2007*; *Ishibashi et al., 2013*), using the SP6 mMessage mMachine kit (Ambion), and final RNA diluted to 50 ng/ml. Equal volumes of the reporter construct(s) and the transposase RNA were mixed immediately prior to injections. 1–2 nl of the solution was micro-injected per embryo and up to 200 embryos were injected at the one- to two-cell stage. Embryos were screened for mosaic fluorescence at 1–5 days post

fertilisation (dpf), that is, 24–120 hpf and raised to adulthood. Germline transmission was identified by outcrossing sexually mature F0 transgenics with Wt fish and examining their progeny for reporter gene expression/fluorescence. 2–3 F0 lines were generated for each construct, and F1 embryos were screened for reporter gene expression driven by the CREs in the transgenic cassette (*Supplementary file 2*). For the landing pad lines, F1 embryos derived from F0 lines showing the best representative expression pattern for the tracking CRE in the cassette were selected for establishing the line, genotyping and confocal imaging (*Figure 1A*). Dual-CRE dual-reporter construct and phiC31 integrase mRNA was injected in one-cell stage embryos from the selected landing line. The injected embryos were observed from 1 to 5 dpf and successful flipping events scored on the basis of loss of tracking CRE-driven reporter gene expression and gain of mosaic eGFP and mCherry expression patterns (*Figure 1B*). This was observed in about 10% of the injected embryos. Embryos with successful integration of the assay cassette were raised to sexual maturity to establish 2–3 independent F0 lines for each CRE pair tested. We observed <5% variability in the reporter gene expression driven by the CREs in F1 embryos derived from independent founder lines (*Supplementary file 2*, *Figure 3A*, *Figure 3—figure supplements 1–4*). A few positive embryos were also raised to adulthood, and F1 lines were maintained by outcrossing. A summary of the number of independent lines analysed for each construct and their expression sites is included in *Supplementary file 2*.

## Mapping of transgene integration site in the landing lines

Transgenic embryos obtained from outcrossing transgenic lines harbouring the landing pad vectors with Wt strain were sorted into eGFP-positive and eGFP-negative groups. The proportion of eGFP-positive embryos were recorded to identify lines with single and multiple independent transgene integration events. Genomic DNA was purified from ~100 eGFP-positive and eGFP-negative embryos derived from outcrossing the transgenic line with potentially single transgene integration event using QIAGEN DNeasy blood and tissue kit (cat no./ID 69504). Ligation-mediated PCR (LM-PCR) (*Dupuy et al., 2005*) was used for mapping the landing pad integration site using previously published protocol (*Davison et al., 2007*). 1 µg of genomic DNA was digested with either NlaIII, BfaI or DpnII and purified using a QIAGEN QIAquick PCR purification kit (cat no./ID 28104). A 5 µl aliquot was added to a ligation reaction containing 150 µmoles of a double-stranded linker. Ligations were performed using high-concentration T4 ligase (NEB, M020S) at room temperature for 2–3 hr. The first round of the nested PCR was performed using linker primer 1 with either Tol2 Left 1.1 or Tol2 Right 1.1 using the following cycling conditions: 94°C (15 s)–51°C (30 s)–68°C (1 min), 25–30 cycles. Second round nested PCR was then performed using linker primer 2 with either Tol2 Left 2.1 or Tol2 Right 2.1 using the following cycling conditions: 94°C (15 s)–57.5°C (30 s)–68°C (1 min), 25–30 cycles. The PCR products were resolved by electrophoresis on a 3% agarose gel, and the products selectively amplified in samples derived from eGFP-positive embryos were cloned and sequenced. Sequences flanking the Tol2 arms were used to search the Ensembl *Danio rerio* genomic sequence database to position and orient the insert within the zebrafish genome. The sequences of the linker oligos and primers used are provided in *Supplementary file 1*.

## Genotyping of transgenic lines bearing dual-CRE dual-reporter constructs

Genomic DNA was isolated from F1 embryos obtained by outcrossing F0 lines established for each assay construct. PCR-based genotyping assay was designed to assess the integration of the cassette in the landing pad. Primers were designed across the junctions of assay vector and landing site (SP1-2, SP11-12) and within the assay cassette (SP3-10). Details of the screening primers (primer sequences and source genome) are provided in *Supplementary file 1*. Genotyping data for a transgenic line described in *Figure 4* are shown in *Figure 4—figure supplement 1*.

## Imaging of zebrafish transgenic lines

Embryos for imaging were treated with 0.003% 1-phenyl2-thio-urea (PTU) from 24 hpf to prevent pigmentation. Embryos selected for imaging were anaesthetised with tricaine (20–30 mg/l) and mounted in 1% low-melting point (LMP) agarose. Images were taken on a Nikon A1R confocal microscope and processed using A1R analysis software. Time-lapse imaging was performed on an Andor Dragonfly spinning disk confocal and processed using Imaris (Bitplane, Oxford Instruments,

RRID:SCR_007370) and Fiji (RRID:SCR_002285). Embryos mounted in 1% LMP were covered with tricaine solution and held in a chamber at 28.5°C.

## Quantification of imaging data

eGFP and mCherry signal intensities were quantified in selected regions of expression in images acquired from F1 transgenic embryos using ImageJ software. Measurements were taken from at least five independent embryos for each line. Mean fluorescence intensity ratios (eGFP/ mCherry, G/C or mCherry/eGFP, C/G) were computed for each expression domain. Average of mean fluorescence intensity ratios was computed using measurements from independent embryos derived from each line for each expression domain and plotted as shown in *Figures 3 and 5*. The level of significance (p-value) of differences in average mean florescence intensity ratios in expressing tissues between different transgenic lines was computed using two-tailed Student's t-test. Raw values of the data plotted are provided in *Figure 3—source data 1* and *Figure 5—source data 1*.

## Distribution of Q-STARZ reagents

All the plasmids required for the assay would be deposited in Addgene, and the landing lines would be made available to the zebrafish scientific community upon request.

## Acknowledgements

This research was funded by a personal fellowship to SB from the Royal Society of Edinburgh/Caledonian Research fund (RSE/CRF personal research fellowship 2014). SB and AM were also supported by a project grant from Newlife Charity for Disabled Children (Grant ref: 17-18/15). WAB was supported by a Medical Research Council (MRC) UK University Unit grant (MC_ UU_00007/2). KU was supported by a PhD studentship from the MRC. ND was supported by the European Union's Horizon 2020 research and innovation program under the Marie Sklodowska-Curie grant agreement no. 642934, Chromatin3D.

## Additional information

### Funding

| Funder | Grant reference number | Author |
|---|---|---|
| Medical Research Council | 632WBI/RH1018 | Shipra Bhatia Kirsty Uttley Wendy A Bickmore |
| Royal Society of Edinburgh | 632WBI/R43399 | Shipra Bhatia |
| Newlife – The Charity for Disabled Children | 632WBI/R45412 | Shipra Bhatia Anita Mann |
| Horizon 2020 | 642934 | Nefeli Dellepiane |

The funders had no role in study design, data collection and interpretation, or the decision to submit the work for publication.

### Author contributions

Shipra Bhatia, Conceptualization, Data curation, Investigation, Methodology, Project administration, Supervision, Writing - original draft, Writing - review and editing; Dirk Jan Kleinjan, Conceptualization, Methodology; Kirsty Uttley, Formal analysis, Investigation, Methodology; Anita Mann, Formal analysis, Methodology; Nefeli Dellepiane, Methodology; Wendy A Bickmore, Conceptualization, Writing - review and editing

### Author ORCIDs

Shipra Bhatia http://orcid.org/0000-0002-2091-7858
Dirk Jan Kleinjan http://orcid.org/0000-0001-6795-3082
Kirsty Uttley http://orcid.org/0000-0003-1929-6705

### Ethics

All zebrafish experiments were approved by the University of Edinburgh ethical committee and performed under UK Home Office license number PIL PA3527EC3; PPL IFC719EAD.

### Decision letter and Author response

Decision letter https://doi.org/10.7554/eLife.65601.sa1
Author response https://doi.org/10.7554/eLife.65601.sa2

---

## Additional files

### Supplementary files

• Supplementary file 1. Details of oligonucleotides used in the study for generation of landing pads and assay constructs, and mapping of site of integration of transgene in landing lines and test lines.

• Supplementary file 2. Overview of transgenic lines generated in the study.

• Transparent reporting form

### Data availability

Source data files contains the numerical data used to generate figures.

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
