## [Decision Letter]

[Editors' note: this paper was reviewed by Review Commons.]

---

## [Author Response]

Reviewer #1:1) It is convincingly shown that adding insulator elements (cHS4) reduces crosstalk between the two PAX6 CREs tested (Figure 3). However, it is unclear if this approach will work for other CREs. This point should be discussed, and perhaps the authors could give some troubleshooting advice (e.g. adding more insulators or trying different insulator elements?). 

The possibility of using more copies of the CHS4 element or another insulator element have been discussed in the revised version of the manuscript (lines 448-454).

2) All CREs used in proof-of-concept experiments in this work have well known activities in zebrafish embryos. A new/uncharacterized CRE has not been tested yet using this system. It is unclear from the workflow (Figure 1B) what happens if the CRE does not drive detectable levels of EGFP/mCherry. How does one determine whether lack of reporter expression is due to technical problem (with the transgene or phiC31 integration) or that the CRE is not active in zebrafish? Perhaps adding a PCR-based genotyping step could address this potential problem? 

A PCR-based genotyping assay is now included in the description of the assay pipeline (lines 612-620) and the genotyping results for one of the transgenic lines is shown in figure 4—figure supplement 1.

3) Other limitations of the system should also be discussed. For example, the system appears to be useful for identifying variant CREs that result in a change (either loss or gain) of temporal or spatial activity, but it is not clear how subtle changes in expression level (either slightly increased or decreased) would be identified or quantified. Perhaps other approaches could be used in combination with this system to fully analyze mutant CRE activity. Another limitation is that this approach is only be applicable to CREs that are active in the first few days of zebrafish embryonic development. 

We have included a section in the discussion describing the potential limitations of our assay (lines 454-458, 492-493)

Minor points: i) Although it is discussed in the previous work published in PLoS Genetics, it is probably worth mentioning here why the gata2 minimal promoter was chosen for the reporter system. 

The choice of the gata2 promoter in our constructs was based on our previously published work. We have re-iterated this and referenced these studies in the workflow description (lines 526-529).

ii) It would be helpful if the cSH4 element is briefly described (e.g. “insulator element”) in Figure 1 legend. 

We have modified the figure legend according to the suggestion.

iii) It is not clear from the manuscript whether the new reagents reported here-including dual reporter vectors and transgenic attB landing site zebrafish strains-will be made available to the scientific community, or how these reagents would be distributed. 

We have included a section describing our plans for distribution of reagents and tools described in the manuscript (lines 649-651). All the vectors would be deposited in Addgene for distribution and all the zebrafish lines would be openly shared with the scientific community.

Reviewer #2: 1. The dual reporter system uses EGFP and mCherry to report the activities of two different CREs in the same animal. However, EGFP and mCherry have drastically different fluorescence properties which have not been measured particularly well in vivo and especially not in zebrafish. They have different maturation times (mCherry is much quicker). Both are quite stable in vivo, but mCherry is particularly stable in cell culture and in vivo, even resisting lysosomal degradation (EGFP does not – it is acid and protease sensitive) (Katayama et al., 2008; McWilliams et al., 2016). Often, promoter activity assays in zebrafish employ short lived "destabilized" FPs, such as destabilized GFP and destabilized dsRed. With stable FPs, false positives could be reported due to the fluorescent signal remaining for a long period of time after promoter activity has ceased. Replacing the traditional FPs with destabilized versions could be one way to improve the temporal resolution of this assay. This is probably not necessary to do in the present study but might be a worthy future direction.

We have discussed the potential use of de-stabalised fluorophores in the Discussion section of the revised version of the manuscript (lines 454-458).

2. However, no matter which pair of FPs is chosen, there will be differences in signal intensity/brightness and decay rate. Thus, the FP swap experiments should be employed for any experiment claiming a temporal (Figure 4) or quantitative (Figure 5) difference between CRE activation or deactivation. If the EGFP/mCherry swap experiments show the same results, the confidence in the assay will be significantly bolstered. We estimate the proposed experiments to take about 4 months to allow for molecular cloning of the FP swapped constructs, injection into the "landing" strain, raising to sexual maturity (2.5 mo), screening for founders, and performing the imaging. These are the only two suggested experiments I would need to feel confident in the results and to recommend publication

We would point out that we included dye-swaps for the PAX6-CREs and the quantification of those data in Figure 3 in the original manuscript. Dye-swap experiment for SBE2WT/SBE2Mut were described in our previous work published in Plos Genetics. However, to increase confidence in our current system we have now also included data from additional dye-swap lines as suggested by the reviewer. These data are included in Figures 4 (SBE2 vs SBE4) and 5 (SBE2 WT vs mutant) and are been described in the Results section of the main text (lines 349-352, 371-373).

Reviewer #3: Major comments 1. First, given the importance of quality landing lines for the methodology, I would like to see more clarity and emphasis on validation of the Shh-SBE2 landing pad in the main text. Based on supplemental tables 1 and 2, this reviewer is somewhat unclear on whether there is one or three lines with Shh-SBE2 based landing pads (one site is mentioned in table 1, but table 3 mentions three F0 lines, and the text is ambiguous). The authors also state that the Shh-SBE2 landing pad is a single copy integration, but the data supporting this conclusion does not appear to be included (linker mediated PCR does not rule out other integrations).

Our first criteria for selecting the landing lines was visualising a clean eGFP expression pattern driven by the tracking CRE included in the landing cassette. The tracking CREs chosen had previously well-characterised CRE activities. As indicated in supplementary file 2, figure 2 and figure 2—figure supplement 1, only the transgenic lines bearing the landing pad with Shh-SBE2 CRE passed this criterion. We screened three independent F0 lines for the Shh-SBE2 landing pad by LM-PCR and transgene segregation analysis. This data supported single site integration in only of the three founder lines, which was subsequently used for all the experiments described in the manuscript. However, we appreciate that this analysis doesn’t rule out multiple tandem integrations of the landing cassette at the described site. Hence, we only refer to the landing pad as ‘single-site integration’ and not as ‘single-copy integration’ in the manuscript text. We have emphasized these details in the revised manuscript text (lines 206-214).

2. It would also be useful to have more clear numbers indicating the reproducibility of the expression pattern in F1 animals. Do 100% of F1 progeny from multiple crosses show the integration show the expression pattern in image 2 A? If there is variability how much, and how many fish were examined? This reviewer also wonders whether appropriate expression of Shh-SBE2 in this landing site is enough to call it neutral. For example, perhaps position effects might be observed with a different weaker CRE in this site? Better documentation will allow for more widespread and appropriate use of the landing pad. 

We do not observe any variability in expression in F1 embryos derived from an individual founder line. This information is now included in the main text file (line no 441-443,565-575) and a representative image of several embryos derived from founder lines for the Shh-SBE2 landing line and all the test lines is now included in figure 3—figure supplement 1-4. Whilst we cannot rule out the possibility of position effects being observed for weaker CREs when integrated in the SHH-SBE2 landing pad, we do not observe any position effects for any of the CREs we have tested in the manuscript (described in supplementary file 2). This is in stark contrast to the previous version of our dual-colour reporter assay described in Bhatia et al. 2015, where we tested some of the CREs described here and observed position effects. We have re-iterated this in the Discussion section.

3. Similar concerns apply to the integration of test constructs. To evaluate the practicality of the approach, it would be useful to have numbers reporting the frequency of recovering F1 individuals with PhiC mediated integration of the reporter into the desired landing site. It is also important to provide better documentation of the degree of reproducibility in expression patterns between F1 progeny. Numbers of embryos imaged and fraction with the indicated expression pattern are needed for all data in the main text. At minimum, gross expression patterns should be examined in at least 10 F1 larvae. If there is variability between individuals, some image documentation of this in supplementary data would be welcome. 

We have included the approximate percentage of successful replacement events in the revised version of the manuscript (line no 224-228, 565-574). As mentioned above, we do not observe variation in expression patterns between F1 embryos derived from individual founder lines and gross expression patterns of F1 embryos for each line have now been included in figure 3—figure supplement 1-4.

Minor comments: i) For figure 1, it may be clearer to present generation of the landing pad lines and screening of CRES using these lines in separated figure panels (B) for generation of landing pads, and (C) for CRE analysis. 

Figure 1 has been modified as suggested by the reviewer

ii) Landing pads that were less effective might also be moved out of figure 2, to the supplemental material to help improve clarity and to allow for focus on the tools with the most utility.

Figure 2 and figure 2—figure supplement 1 have been modified as suggested by the reviewer. Figure 2 now describes data from the landing line subsequently used in all the experiments described in the manuscript.

iii) Scale bars should be included in all images, 

We have now included scale bars in all the images.

iv) In some cases, image labeling somewhat obscures the relevant features 

All figures have been modified to rectify this

v) To help evaluate consistency, in all relevant figures (4, 5, sup Figure 3 etc) the number of embryos examined should be included in the legend. 

This information has now been included in the figure legend.